# Functional and Mechanical Properties of As-Deposited and Heat Treated WAAM-Built NiTi Shape-Memory Alloy

**Arthur Khismatullin** [1,*], **Oleg Panchenko** [1], **Dmitry Kurushkin** [1], **Ivan Kladov** [1] **and Anatoly Popovich** [2]

1   Laboratory of Lightweight Materials and Structures, Institute of Machinery, Materials, and Transport, Peter the Great St. Petersburg Polytechnic University, 29 Polytechnicheskaya St., 195251 St. Petersburg, Russia; panchenko_ov@spbstu.ru (O.P.); kurushkin_dv@spbstu.ru (D.K.); Kladov.Iv.Vl@yandex.ru (I.K.)
2   Institute of Machinery, Materials, and Transport, Peter the Great St. Petersburg Polytechnic University, 29 Polytechnicheskaya St., 195251 St. Petersburg, Russia; director@immet.spbstu.ru
*   Correspondence: hismat_ar@spbstu.ru

**Abstract:** In this work, MIG process was utilized for the wire arc additive manufacturing of the wall-shaped parts, using NiTi shape-memory alloy. High-scale specimens consisting of 20 layers were deposited by using Ni-rich (Ni55.56Ti wt.%) wire as a feedstock on the NiTi substrate with the identical chemical composition. One of two specimens was heat-treated at a temperature of 430 °C for 1 h. The influence of such a heat treatment on the microstructure, phase transformation temperatures, chemical and phase compositions, microhardness, and tensile and bending tests' results is discussed. As-deposited metal successfully demonstrates superelastic behavior, except in the lower zone. In regard to the shape-memory effect, it was concluded that both the as-deposited and the heat-treated samples deformed in liquid nitrogen completely restored (100%) their shapes at an initial strain of 4–5%. An occurrence of the R-phase was found in both the as-deposited and the heat-treated specimens. The phase transformation temperatures, microstructure, and tensile and bending tests results were found to be anisotropic along the height of the specimens. The presented heat treatment led to changes in the functional and mechanical properties of the specimen, provided with the formation of finely dispersed Ni4Ti3, NiTi2, and Ni3Ti phases.

**Keywords:** wire arc additive manufacturing; shape-memory effect; gas metal arc welding; Nitinol



## 1. Introduction

There is a class of materials that have a shape-memory effect. Such a group has gained interest from industry and scientists over the years due to its superior properties and the possibility to develop new automotive, aerospace, medical, and robotics applications. This group includes a wide variety of materials [1–5], but NiTi-based alloys have the best combination of characteristics compared with the others [6]. For the first time, in 1963, Buhler et al. discovered the unique shape-memory effect of a close-to-equiatomic NiTi alloy [7]. Among other applications, NiTi alloy parts have found their application in various industries as motors, drive mechanisms, actuators, various types of medical implants, etc. [8].

The main problem of NiTi parts production is the manufacturing costs [8,9]. In order to deal with this problem, new manufacturing methods are studied and applied. Nowadays, more and more conventional manufacturing techniques are reaching their technological limits. In addition, current trends in the reduction of material waste, together with an increase in production efficiency, lead to the emergence of new technological processes. One of such a new family of methods is additive manufacturing (AM). A lot of research has already focused on the improvement of AM process efficiency, not only for conventional alloys (e.g., stainless steels, aluminum alloys, etc.), but for functional and smart materials such as shape-memory materials (specifically NiTi alloy). One of those research studies

was conducted by Mahyar Khorasani [10]. In their work, authors not only observe current AM techniques but also discuss different optimization approaches in design, including topology optimization, support optimization, and selection of part orientation and part consolidation. The paper highlights the ways to reduce the production cost of manufacturing complex details in the aerospace industry, using AM techniques. This case study presents a detailed discussion on the comparison of conventional techniques with the AM. The economic superiority of the laser-based powder bed fusion AM technique in comparison with mechanical processing is shown numerically, using the example of manufacturing an air manifold. In regard to that, there is a lot of other interesting information, and the reader is invited to read the full text of the article at the link in the description.

Powder-based techniques for AM of NiTi parts are used widely [11–15]. One of the common techniques for additive manufacturing of NiTi shape-memory alloy is selective laser melting (SLM). There are lots of relevant research studies in this field. An example is the work of H. Z. Lu et al. [16], in which the authors discussed the laser power and scanning speed, as well as their influence on the phase composition and transformation temperatures. In addition, much attention has been paid to bending shape-memory properties. Optimal process parameters provide a tensile strength of 788 MPa and an elongation of 7.43% in the austenitic state. The results of the study demonstrate SLM for NiTi parts as a promising technology for industrial applications. Another research study on optimization of SLM process parameters was performed by Zhenglei Yu et al. [17]. The paper presents a detailed study of the mechanism of the influence of process parameters on the microstructure, mechanical properties, corrosion resistance, and friction resistance of the NiTi alloy that was fabricated by using SLM. The main parameters analyzed were laser power and scanning speed. For more detailed information, the reader is invited to refer to the full version of the article, a link to which is provided in the references. The work of Jian-Bin Zhan, again, focused on the process parameters of SLM NiTi [18]. That fact confirms the relevance of this topic. A relative density of approximately 99% was obtained in the power range of 45–60 W. It was found that the martensite transformation temperatures increased with the energy density. However, the authors conclude the increasing of defects, together with higher energy densities. Interesting discussions on this topic are also given in the article.

Nowadays, there is another new and promising additive manufacturing method called Wire and Arc Additive Manufacturing (WAAM), which was used for the study in this paper. The use of metal wire as a filler material instead of powder leads to a reduction in material waste and manufacturing cost, but it also reduces the complexity of the built part [19,20].

The first published study of WAAM technology using NiTi shape-memory alloys (SMA) was introduced by Wang et al. in 2019 [21]. The authors of that study successfully confirmed the feasibility of in situ synthesis of Ni-rich NiTi alloys by using a double-wire-feeding WAAM process with Ni and Ti wires. The phase composition and functional properties of the built metal were found to be anisotropic. Later, in 2020, the same team performed a new series of experiments for the evaluation of crystallographic orientation change, precipitation processes, phase transformation, and mechanical properties with increased current for the dual-wire arc additive manufacturing of Ni-rich NiTi alloy parts [22]. The research was focused on the optimization of the WAAM process parameters for the fabrication of NiTi components. In 2021, Wang et al. published the third paper [23]. It was focused on the characterization of changes in crystallographic orientation, phase transformation, and mechanical properties depending on the preheating temperature of the substrate.

Another research group led by Zeng published their work in 2020 [24]. It was focused on the demonstration of superelastic behavior under tensile conditions, highlighting the potential of applying WAAM to the printing of NiTi complexly shaped parts. Superelasticity under tensile conditions after seven load/unload cycles was demonstrated in that paper.

Another important work was performed by Resnina et al. in 2021, in which a five-layered specimen was successfully deposited [25–27]. Phase composition, precipitation formation, transformation temperatures, microstructure, and mechanical properties were

discussed in detail for each layer of material. Particular attention was given to recoverable strain variation upon cooling and heating under stress. The results show that the martensitic transformation occurs at different temperatures in the layers due to anisotropy of the chemical composition in metal. Thus, the recoverable strain variation upon cooling and heating under a stress occurs by stages. Moreover, the authors showed a typical microstructure formation mechanism for WAAM, explained the correlation between microstructure and properties, and discussed the influence of various heat treatments on the microstructure and properties of the material.

The influence of technological parameters (arc voltage) on the microstructure and functional properties were discussed by Ponikarova et al. [28]. A cold metal transfer process (metal inert gas welding) was used for the NiTi metal deposition, and an evaluation of the microstructure and mechanical properties was performed by Lin Yu et al. [29]. In 2020, Shen et al. published research on the in situ neutron diffraction study on the high-temperature thermal phase evolution of WAAM NiTi [30].

To sum up this literature review, the following should be mentioned: all the discussed studies, except for Reference [30], used very small specimens that consisted of five layers on average. Since WAAM is a high-scale parts' printing method, it is of practical interest to deposit a macroscopic metal object—a wall with a size comparable to those in other papers dedicated to WAAM-built metal study. Hence, the evaluation of the microstructure and mechanical properties of a such metallic part is also of scientific interest. In almost all of the studies, except for References [25,28,29], the tungsten inert gas approach for deposition was used, so the feasibility of the metal inert gas (MIG) process for the deposition of a macroscopic object should be explored. It is well established in the discussed papers that all of the additive technologies produce metallic parts with properties anisotropy in comparison with the conventional manufacturing types, which typically produce isotropic materials. A much smaller number of topics are focused on the influence of heat treatment on the anisotropic NiTi alloy, as fabricated by wire and arc additive manufacturing. Thus, in this paper, heat treatment is also addressed.

The aim of this study was the investigation of WAAM–MIG deposition of a macroscopic specimen made of NiTi shape-memory alloy. A relatively large-size specimen that consists of several layers would be discussed from the view of the microstructure, mechanical and functional properties, phase composition, and its changes under the influence of one of the common heat treatment methods. Moreover, the feasibility of the MIG process for WAAM deposition of a NiTi shape-memory alloy is discussed.

## 2. Materials and Methods

In this investigation, a wire with a diameter of 1.2 mm Ni-rich (55.56 wt.%) was used as a filler material. The substrate used as a baseplate for deposition was also Ni-rich (55.56 wt.%) NiTi alloy. MIG process was used for WAAM of wall-shaped parts; no additional synergetic technologies, such as "CMT" or others, were used. Deposition was performed in a controlled atmosphere chamber filled with Ar (99.99% pure) to prevent oxidation. The WAAM system consisted of Motoman MH24 robot (Yaskawa) and EWM Alpha Q 552 welding power source. The process parameters were chosen to achieve maximum process stability. For this reason, the values were gradually changed to the best combination with minimal spatter, effective gas shielding, and stable metal transfer to the weld pool. The resulting WAAM process parameters were as follows: the torch travel speed was 35 cm/min, wire feed rate was 4.5 m/min, deposition average current was 196 A, and arc average voltage was 13.0 V. Tescan Mira3N scanning electron microscope (TESCAN, Brno, Czech Republic) equipped with energy dispersive spectroscopy module (EDS) was used to study the microstructure and chemical composition. Phase-transformation temperatures were studied by a differential scanning calorimetry (DSC) on a DSC 204 F1 machine (NETZSCH-Gerätebau GmbH, Selb, Germany). DSC samples had the shape of a cube with parameters of $3 \times 3$ mm. Degree of reversible deformation was evaluated by using micrometer and self-made bending machine. The samples for bending tests had the form of

a thin strip with parameters $50 \times 2 \times 1$ mm directed along the layers. The determination of the shape-recovery degree was carried out by the thermomechanical method. The samples were deformed up to 10%, unloaded, and heated, fixing all the parameters of the shape changes. The degree was defined as the ratio of reversible strain to induced strain. The tensile tests were performed by using dog-bone-shaped samples with the following parameters: total length of 35 mm, work length of 18.18 mm, gripper diameter of 7 mm, and work diameter of 3 mm. Tensile tests were performed on a Tinius Olsen Super "L" 120 testing machine. X-ray diffraction (XRD) phase analysis was performed by using a Bruker D8 Advance setup. Samples for the experiments were extracted from the specimens by using an electrical discharge machine, Meatec DK7763ME12 (MEATEC, Moscow, Russia). The average grain size was estimated by the intersection method: three random straight lines were drawn through the horizontal and vertical directions of the micrographs; the number of grain boundaries crossing the lines was counted; the length of the lines was divided by the number of intersections, and the obtained values were taken as the specific (width or height) average grain sizes. In order to investigate the influence of heat treatment on the microstructure, phase composition, mechanical properties, and functional properties of the as-deposited metal, aging treatment was utilized at a temperature of 430 °C for 1 h; it corresponds to the aging processes in the Ni-rich TiNi alloys [31].

## 3. Results and Discussion

### 3.1. Macrostructure and Microstructure

Two specimens in the form of a wall consisting of 20 layers were deposited. The direction of deposition was changed after each layer. The walls' geometry was as follows: length—110 mm, width—10 mm, and height—38 mm. Figure 1 represents the as-deposited wall (a) and a schematic illustration of the samples' position for the study inside both specimens (b). A second wall was used to investigate the influence of the heat treatment.

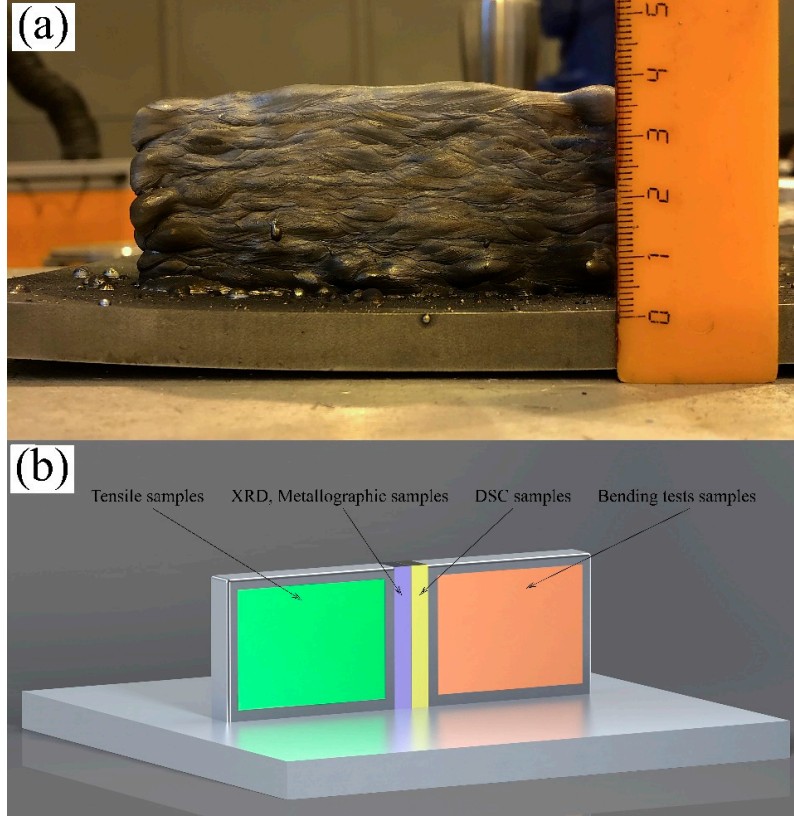

**Figure 1.** As-deposited wall (**a**) and schematic representation of the zones for samples extraction from the wall (**b**).

The wall surface's appearance indicates the feasibility of the MIG process utilization for the near-net-shape objects' printing. Nevertheless, it should be mentioned that the arc wandering effect was observed during printing. The abovementioned phenomenon was similar to that, occurring in the WAAM process, using Ti-alloy wire [32], and it affects the quality of the surface.

A panoramic image of the microstructure is shown in Figure 2. It should be noted that metal incorporates some small pores. The specimen has three major zones: a zone close to the substrate with large (350 μm in width and 1400 μm in height) columnar grains (Figure 2 (Zone 3)); a middle zone with smaller (270 μm in width and 1200 μm in height) columnar grains compared with the third zone (Figure 2 (Zone 2)); and the upper zone with the smallest columnar grains (250 μm in width and 1050 μm in height), which has small equiaxed grains on the top (Figure 2 (Zone 1)). Columnar grains are observed in the microstructure along the entire height of the deposited specimen to be crossing clearly visible fusion lines; thus, grains were formed during epitaxial growth. The crystallization front is directed from the lower layers to the upper ones. Since most of the volume of the specimen exhibits columnar grain growth, all the samples for the following studies were cut out of the metal with this microstructure.

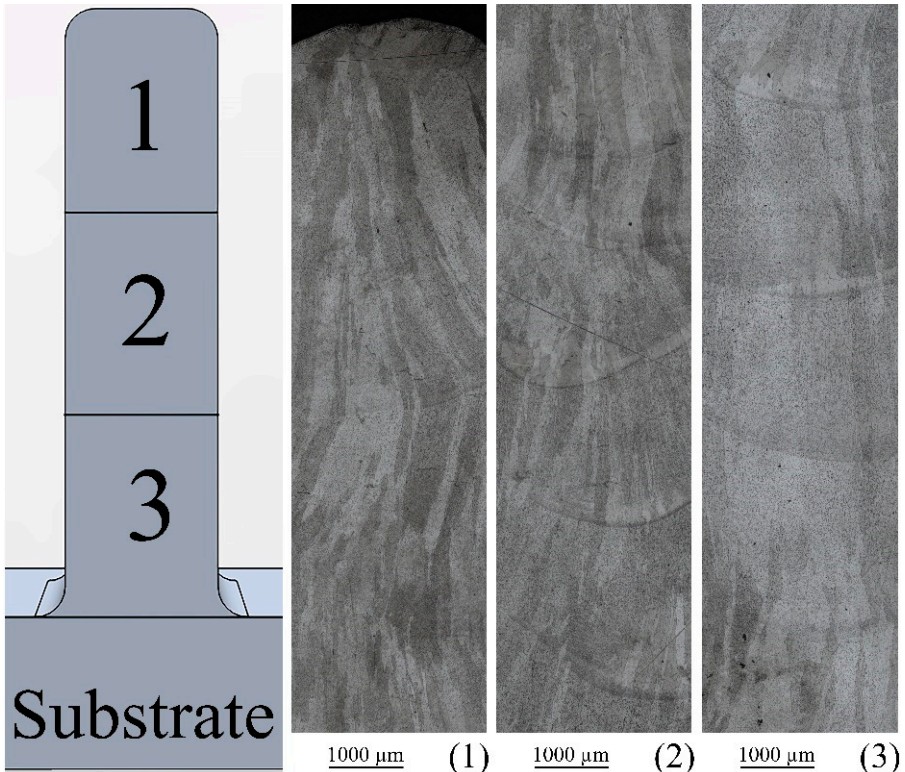

**Figure 2.** Macrostructure of NiTi WAAM-built metal.

The abovementioned microstructure characterization also fits with the one observed with higher magnification obtained with SEM in Figure 3a–c.

Observations with optical microscopy of the heat-treated specimen (see Figure 4) show a fine martensitic microstructure with a high angle of disorientation [31].

The SEM microstructure analysis of the heat-treated metal (Figure 5) shows the formation of new phases; for example, near-globular-shape dark points in the microstructure are either the traces of precipitates grinded during the metallographic preparation process or are precipitates themselves. Similar results were obtained in Reference [30]. On the basis of the literature review on the heat treatment, it is believed that the precipitates are Ti2Ni [31]. Phase composition are discussed further in Section 3.4.

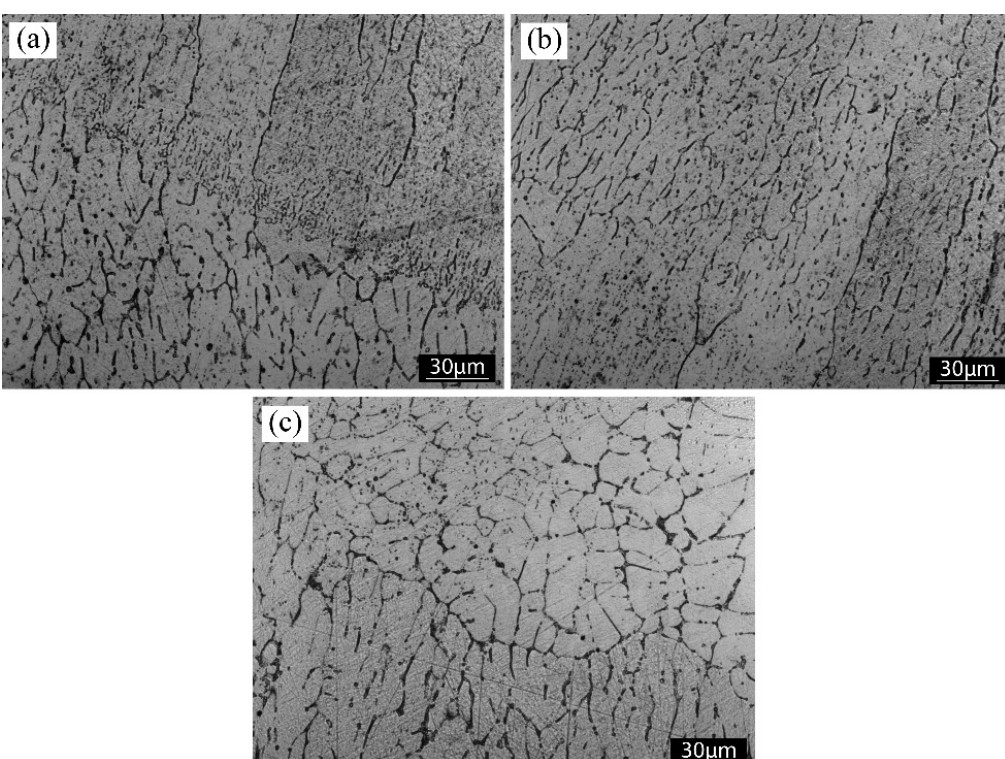

**Figure 3.** Microstructure of the typical (**a**) fusion line between the layers, (**b**) columnar grains observed along the entire structure, and (**c**) equiaxed grains visible on the top.

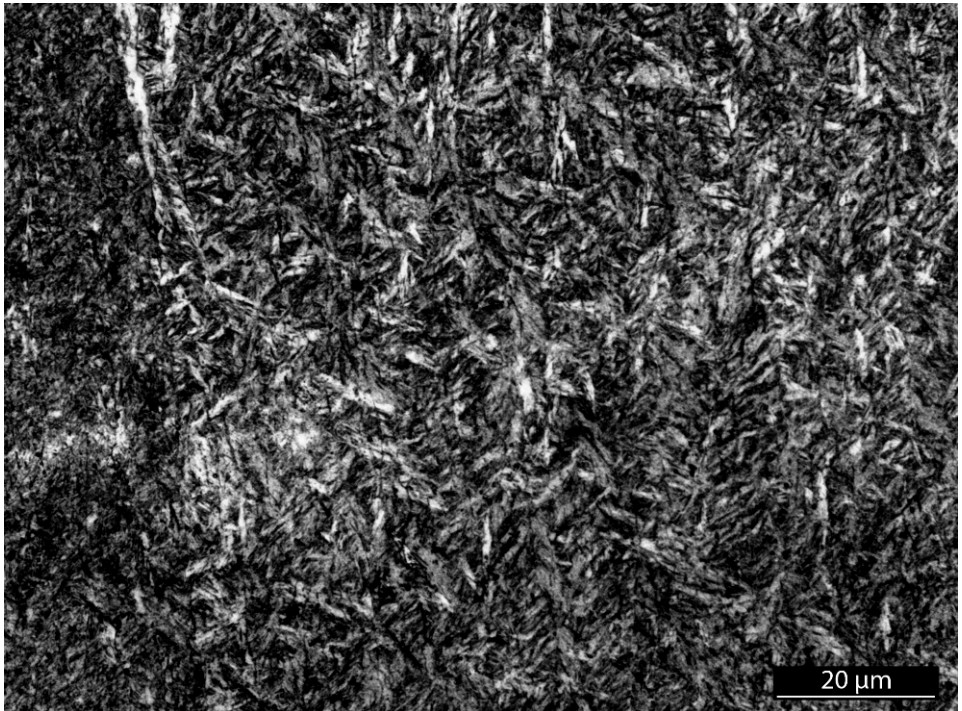

**Figure 4.** Microstructure of the heat-treated metal.

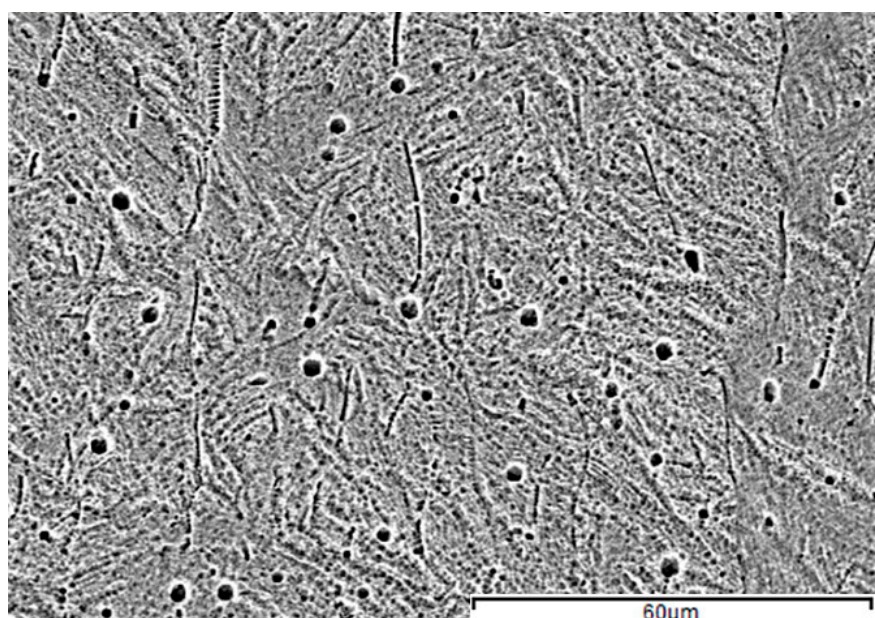

**Figure 5.** SEM image of the metal after heat treatment.

*3.2. Chemical Composition and Phase Transformation Temperatures Determination*

To assess the differences in chemical composition between different areas of the manufactured specimen, an EDS characterization was carried out.

It was possible to establish a change of Titanium content in the space between equiaxed grains (Figure 6a, Spectrum 1), i.e., is 51.52 at.%; in the grains (Figure 6a, Spectrum 2), it is 50.51 at.%. The titanium content in the fusion line between the layers (Figure 6b, Spectrum 1) is 50.55 at.%, while in the metal of the layer (Figure 6b, Spectrum 2), it is 50.16 at.%. These facts could be explained by the difference in solidification temperature of Ti and Ni, leading to liquation processes during deposition.

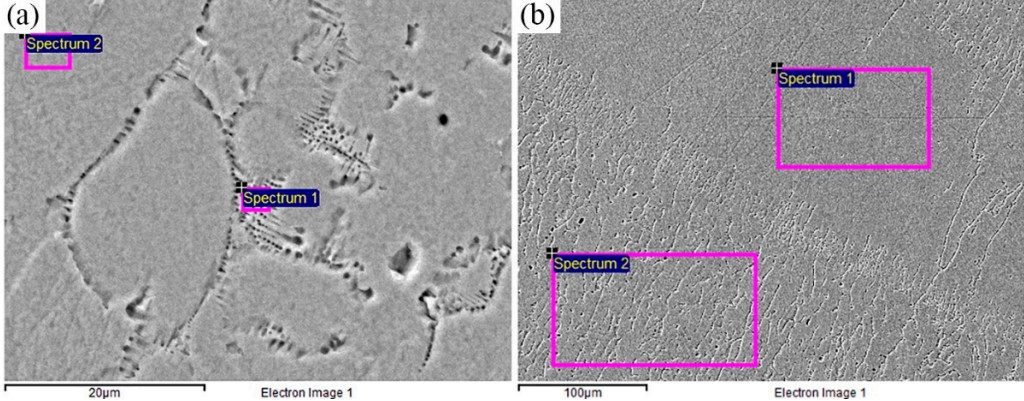

**Figure 6.** EDS characterization of (**a**) equiaxed grains; (**b**) fusion lines and layers.

To establish a general trend in the chemical composition change, EDS analysis was performed in all three regions (Figure 2) along the vertical axis of the as-deposited and heat-treated specimens. The results show near-homogeneous distribution of Ni atoms along the height of the walls, without significant differences between the as-deposited and heat-treated specimens. The average Ni content for the three discussed zones was 50.56 at.% for the lower zone, 50.54 at.% for the middle zone, and 50.43 at.% for the upper zone of the specimens.

Based on the EDS analysis, it should be concluded that no relevant anisotropy in the content of chemical elements can be observed.

The determination of the characteristic temperatures of forward and reverse martensitic transformations (MTs) was carried out on samples from the lower, middle, and upper zones of the as-deposited and heat-treated specimens (Figure 7) via DSC. However, for anisotropy characterization, in the as-deposited specimen, the samples for DSC were cut out from central part of manufactured wall and from the edge.

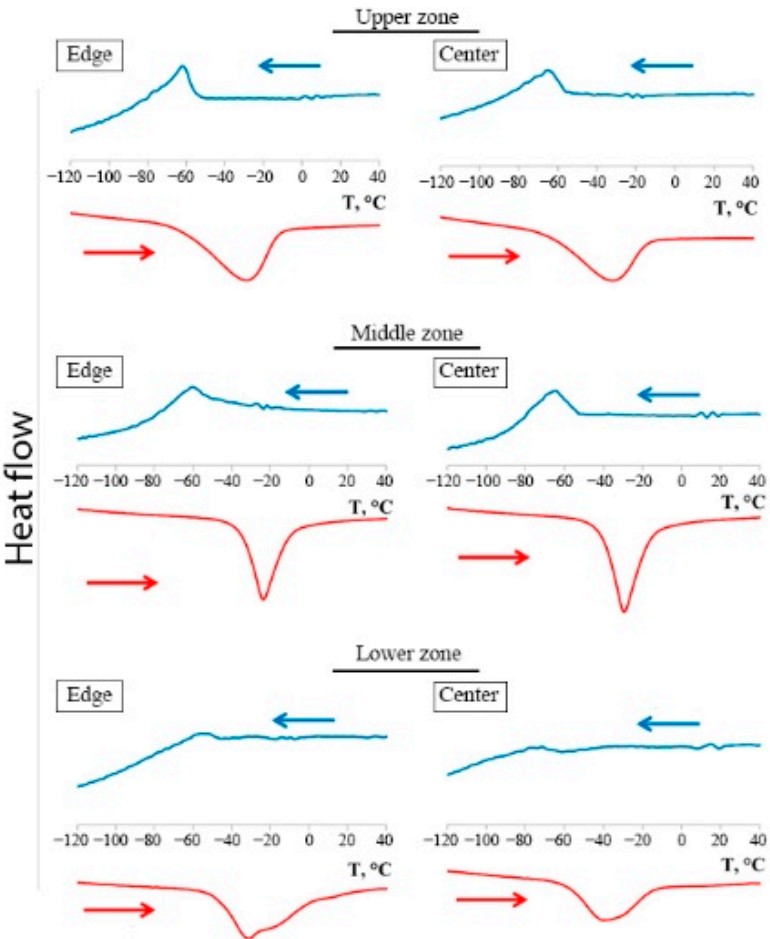

**Figure 7.** DSC curves for samples cut from different zones of the wall.

The original calorimetric curves (DSC) of the samples cut from different zones of the specimen are shown in Figure 7, grouped by zones: on the left are DSC curves related to the edge of the specimen, from the right to the central axis of the specimen.

The results of determining the characteristic temperatures of the phase transformations for all investigated samples are presented in Table 1.

**Table 1.** DSC results for 3 zones of the specimen, °C.

| Zone | $Ni_{av}$, at.% | $M_s$ | $M^{max}$ | $M_f$ | $M_s$–$M_f$ | $A_s$ | $A^{max}$ | $A_f$ | $A_f$–$A_s$ | $A^{max}$–$M^{max}$ |
|---|---|---|---|---|---|---|---|---|---|---|
| Upper zone, edge | 50.43 | −56 | −62 | −71 | 15 | −58 | −28 | −13 | 45 | 34 |
| Upper zone, center | | −52 | −65 | −75 | 23 | −60 | −32 | −15 | 45 | 33 |
| Middle zone, edge | 50.54 | −43 | −60 | −73 | 30 | −36 | −24 | −13 | 23 | 36 |
| Middle zone, center | | −56 | −64 | −80 | 24 | −40 | −30 | −16 | 24 | 34 |
| Lower zone, edge | 50.56 | −43 | −55 | −98 | 55 | −48 | −31 | −4 | 44 | 51 |
| Lower zone, center | | −62 | −78 | −107 | 42 | −57 | −40 | −19 | 38 | 59 |

Ms—martensitic transformation start temperature; $M^{max}$—martensitic transformation peak temperature; Mf—martensitic transformation finish temperature; As—austenitic transformation start temperature; $A^{max}$—austenitic transformation peak temperature; Af—austenitic transformation finish temperature; $Ni_{av}$—average Ni content.

Depending on the investigated zone of the specimen (edge or center) for the upper and middle zones, the peaks of the forward and reverse MT slightly shift (3–6 °C) toward lower temperature values (Figure 7). The main differences for the lower zone in comparison with the middle and upper ones are that Mf temperatures decreased from nearly −70, −80 °C to −107 °C; and an increase of Ms–Mf and $A^{max}$–$M^{max}$ intervals from 23 to 42 and from 33 to 59, respectively, shows the wider temperature hysteresis of martensitic transformation (Table 1).

Observation reveals drastic change of the colorimetric curve for the metal of the lower zone in comparison with the middle and upper zones (Figure 7). Thermal cycles should be considered as the reason for these changes. The lower part of the metal (lower zone) was exposed to repeated heating and cooling during each layer deposition process. This type of heat treatment can be classified as a partial aging due to a relatively low temperature and treatment time. The colorimetric curve of the lower central zone shows an intermediate microstructure evolution step between the metal of the upper zone (where several layers are deposited) and the heat-treated metal (Figure 8).

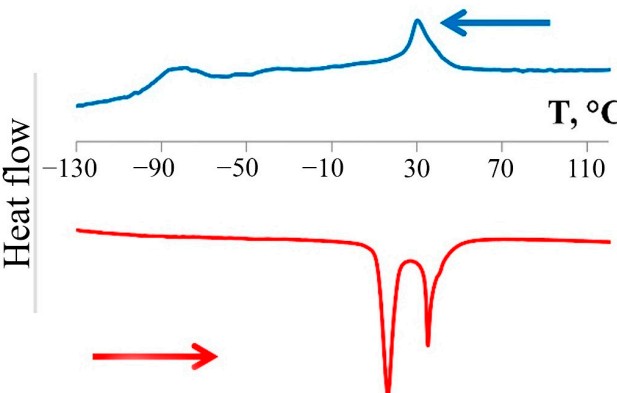

**Figure 8.** DSC curves for heat-treated specimen.

The metal of the middle zone has a narrow interval of reverse MT, which is also more typical for the heat-treated NiTi [33]. The start of two peaks' formation on the colorimetric curve can be observed for the reverse MT in lower zone, as evidence of the R-phase formation [6].

The DSC curves of the heat-treated specimens are shown in Figure 8. The results of determining the characteristic temperatures of the phase transformations are presented in Table 2.

**Table 2.** DSC results from the middle zone of heat-treated specimen, °C.

| Zone | $R_s^f$ | $R_{max}^f$ | $R_f^f$ | $M_s$ | $M^{max}$ | $M_f$ | $R_s^r$ | $R_{max}^r$ | $R_f^r$ | $A_s$ | $A^{max}$ | $A_f$ | $R_s^f$–$M_f$ | $R_s^r$–$A_f$ |
|---|---|---|---|---|---|---|---|---|---|---|---|---|---|---|
| Middle zone, center | 40 | 30 | 25 | −78 | −80 | −99 | 11 | 16 | 20 | 34 | 35 | 38 | 139 | 27 |

$R_s^f$—R-phase forward transformation start temperature; $R_{max}^f$—R-phase forward transformation peak temperature; $R_f^f$—R-phase forward transformation finish temperature; $R_s^r$—R-phase reverse transformation start temperature; $R_{max}^r$—R-phase reverse transformation peak temperature; $R_f^r$—R-phase reverse transformation finish temperature.

As can be seen from Figure 8, the transformation becomes a fully two-stage B2→R→B19′ and goes through the R-phase [6]. B2 → R transformation is in the region of positive temperatures, the main R → B19′ transformation temperature decreased for ~30 °C in comparison with the initial state (see Figure 8). The temperature interval of the direct MT became wider. The temperature of the onset of the B2 → R transformation is 40 °C, while the end temperature of direct MT is −99 °C. The reverse MT takes place in a much narrower temperature range: from 11 to 38 °C. It can be assumed that these changes are

associated with the active release of the Ti3Ni4 phase particles as a result of the aging processes. Internal stress fields from Ti3Ni4 particles stimulate B2 → R transformation and prevent growth of the B19′-martensite plates [33].

### 3.3. Bending Tests

The observed variability of phase transformation temperatures across the WAAM-built metal zones indicates anisotropy that should also influence the mechanical properties.

According to DSC results, the metal of the as-deposited wall is in the austenitic phase at room temperature. Thus, in order to investigate the value of strain recovered due to the superelasticity effect ($\varepsilon_{se}$), we performed bending tests for the metal of the as-deposited specimen at a room temperature, with the initial strain ($\varepsilon_i$) of 5.8% in all three zones. The initial strain value was calculated as a ratio of deformed state to initial state according to micrometer data. The resulting effect of superelasticity was measured as the ratio of the final state to the initial state. Tests results are as follows: upper zone, 92% recovery; middle, 95%; and lower zone, metal fractured. Thus, as-deposited metal successfully demonstrates superelastic behavior except of lower zone. This fact is associated with heterogeneity of microstructure and properties along the vertical axis of the wall-shaped specimen.

In order to investigate the value of strain recovered due to the shape-memory effect, bending tests for the as-deposited specimen at the boiling point of liquid nitrogen (−196 °C) were performed. After the deformation, the samples were heated up to room temperature. All the measurements were performed identically to the bending tests for superelasticity investigation. The results are presented in Table 3. It was concluded that the samples that were deformed in liquid nitrogen completely (100%) restored their shapes at an initial strain of 4–5%, while samples with an initial strain of 6.9–7.8% restored only partially (96–99%). An increase in the initial strain up to 7.9% led to the fracture of the samples from the upper zone.

**Table 3.** The results of the bending tests during the investigation of the shape-memory effect for each zone of the as-deposited specimen.

| Zone | Initial Strain, % | $\varepsilon_{sme}$, % |
|------|------|------|
| Upper zone | 5 | 100 |
| | 7 | 99 |
| | 7.9 | fractured |
| Middle zone | 4 | 100 |
| | 7 | 96 |
| | 7.8 | 97 |
| Lower zone | 4.9 | 100 |
| | 6.9 | 98 |
| | 7.7 | 97 |

$\varepsilon_{sme}$—strain, recovered due to shape-memory effect (SME).

The heat-treated specimen was also subjected to the bending tests at the boiling point of liquid nitrogen. The resulting values are presented in Table 4. Aging treatment at 430 °C for 1 h leads to specimen failure at an initial strain of 6.4 %.

**Table 4.** The results of the bending tests during the investigation of the shape-memory effect for the middle zone of the heat-treated specimen, %.

| Zone | Initial Strain, % | $\varepsilon_{sme}$, % | Irrecoverable Strain, % |
|------|------|------|------|
| Middle zone | 3.2 | 100 | 0 |
| | 4.6 | 100 | 0 |
| | 6.4 | Fractured | Fractured |

### 3.4. XRD Analysis

In order to determine the phase composition of the obtained specimens and investigate its changes after heat treatment, the specimens were subjected to an X-ray analysis. The as-deposited specimen's and the heat-treated specimen's phase compositions were studied; the results are presented in Figure 9. The X-ray diffraction data show that the as-deposited specimen is presented not only by the B2 matrix but with some amount of B19′ phase. This fact can be associated with the stress-induced MT as a result of residual stress generated during the deposition process [23]. In regard to aging at 430 °C, 1 h provides the formation of a finely dispersed Ni4Ti3, NiTi2, and Ni3Ti phases and changes the prevailing phase from a previously B2 + B19′ combination to B19′ + R. The formation of the R phase instead of the B2 phase is confirmed with the DSC results (see Figure 7) for the heat-treated specimen. It should be noted that the retained B2 phase in the XRD results was not observed in the heat-treated specimen. This fact can be associated presumably with an extremely low amount of the B2 phase and the overlapping of the different phases' peaks.

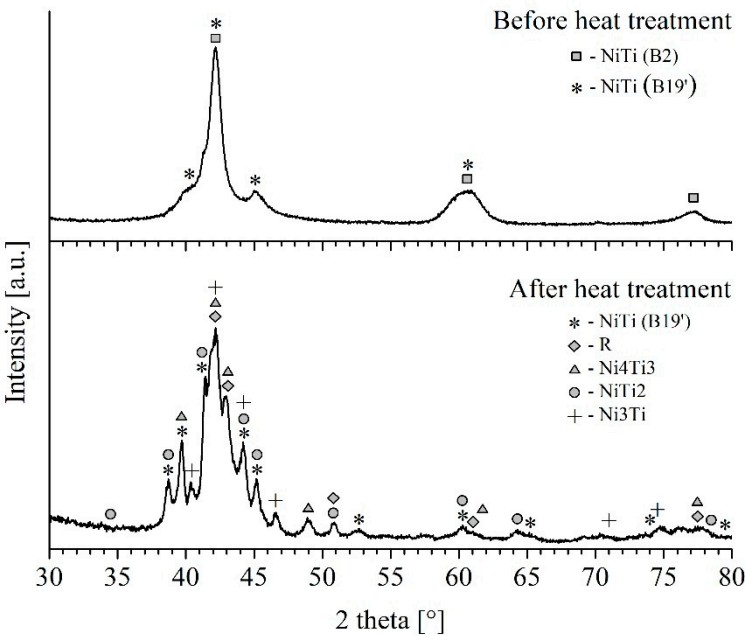

**Figure 9.** X-ray diffraction pattern before and after heat treatment.

### 3.5. Mechanical Properties

The microhardness measurements were performed along the vertical direction in the middle of the wall. The results for non-heat-treated and heat-treated specimens are presented in Figure 10. A decrease in hardness of the heat-treated metal is clearly observed in all of the WAAM metal zones; such an effect was also observed by Lin et al. [29].

The average hardness values along the high of the specimen decreased with the aging: lower zone, from 382 to 347 HV; middle zone, from 370 HV to 348 HV; and upper zone, from 349 HV to 330 HV. According to the literature, there is an inverse relationship between the hardness and the Ms temperature [34]. Moreover, the DSC results, which were presented earlier in this study, show that the Rs temperatures (as a start of martensitic transformation [6]) were suppressed to the direction of positive temperatures after heat treatment. The higher Ms temperature resulted in the lower hardness of the aged Ni-rich NiTi alloy [34].

The mechanical properties of the WAAM-built samples before and after heat treatment were evaluated by conventional tensile tests of six samples (three as-deposited samples and three heat-treated samples) at room temperature. The resulting stress–strain curves of each specimen are depicted in Figure 11a–f.

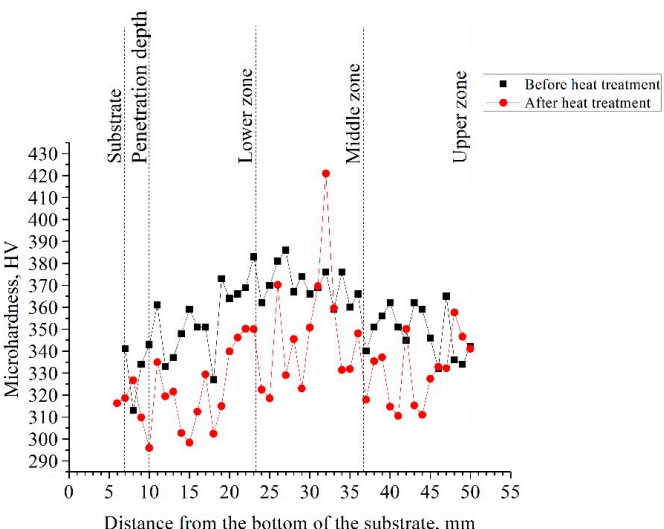

**Figure 10.** Microhardness measurements of the as-deposited and heat-treated specimens.

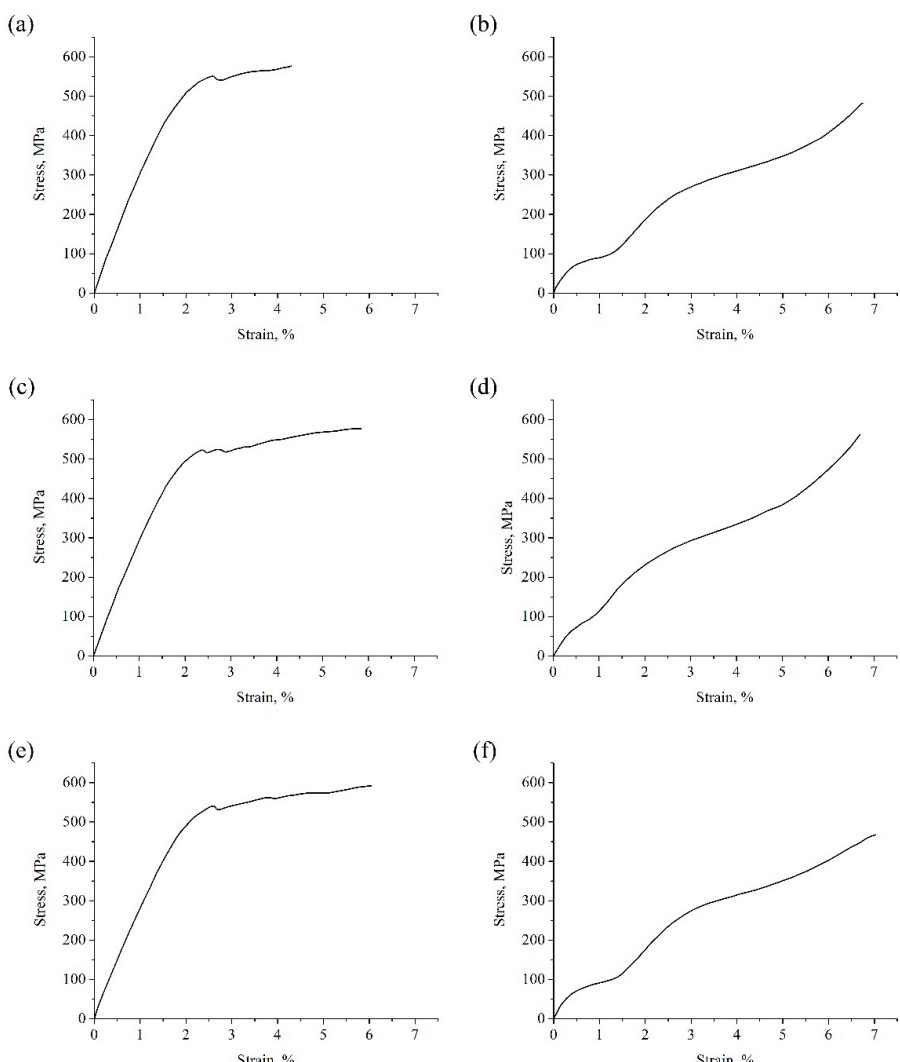

**Figure 11.** Tensile stress–strain curves of the NiTi WAAM samples: (**a**,**c**,**e**) as-deposited samples and (**b**,**d**,**f**) heat-treated samples grouped for three zones: (**a**,**b**) upper zone, (**c**,**d**) middle zone, and (**e**,**f**) lower zone.

The resulting mechanical characteristics are similar to the results of other groups (Table 5) [21,25]. The results show notable changes in the shape of the stress–strain curves after the heat treatment. There is some non-linear zones of the stress–strain curves for the heat-treated samples around 100–150 MPa (Figure 11b,d,f), presumably, influenced by the stress-induced B2–R transformation, after which fully R-phase metal matrix deformed. As it was concluded from the DSC results (see Figure 8), the heat-treated metal matrix at room temperature consists of the R and B2 phase. The XRD results also confirmed this fact (Figure 9). Thus, during heat-treated samples' deformation, two stress-induced transformations occur, namely B2 → R and R → B19′, that alter the stress–strain curve shape.

**Table 5.** Mechanical properties of the as-deposited and heat-treated specimens.

| Specimen | Zone of the Specimen | Ultimate Tensile Strength (UTS), MPa | Elongation, % |
|---|---|---|---|
| As-deposited specimen | Upper | 576 | 4.3 |
| | Middle | 580 | 5.8 |
| | Lower | 592 | 6.0 |
| Heat-treated specimen | Upper | 482 | 6.7 |
| | Middle | 561 | 6.7 |
| | Lower | 467 | 7 |

A decrease in the ultimate tensile strength of the heat-treated specimen in comparison with the as-deposited one is associated with the observed Ni4Ti3 precipitates. Ni4Ti3 particles induce additional internal stress, which suppresses the martensitic transformation [33].

## 4. Conclusions

In this study, high-scale NiTi alloy walls were successfully produced by using the metal inert gas process, confirming the feasibility of MIG deposition for WAAM of shape-memory alloy parts.

The following can be concluded:

1. The microstructure of the as-deposited specimen for the chosen wire chemical composition (55.56 wt.% Ni) is austenitic and mostly consists of columnar grains almost through the entire height of the specimen. There is a small zone in the upper part that exhibits a different microstructure morphology—fine equiaxed grains. Nevertheless, the microstructure of the as-deposited wall was found to be austenitic on the basis of optical and scanning electron microscopy. Evidence of the R-phase formation in the lower zones was found by utilizing the DSC.

2. The drastic variability of the DSC curves and bending tests results for the walls metal in different zones demonstrates the anisotropic behavior of the printed NiTi metal. The main differences in the phase transformation temperatures are a shift of the Mf temperature from −75 °C in the upper part of the wall to −107 °C in the lower part, and an increase of Ms-Mf and $A^{max}$–$M^{max}$ intervals from 23 to 42 and 33 to 59, respectively. Layer-by-layer deposition was found to affect the lower zones of the NiTi walls as the heat treatment similar to aging. The DSC curves of the lower zones of as-deposited specimen were found to be an intermediate between those of the upper zones and heat-treated ones (at 430 °C for 1 h).

3. Aging at 430 °C for 1 h led to the shift of the functional and structural properties due to microstructural and phase transformations: morphology became martensitic (B19′), and the R-phase was formed together with Ni4Ti3, NiTi2, and Ni3Ti. The microhardness and tensile properties of the heat-treated metal decreased (UTS decreased from 576 down to 467 MPa). Stress-induced phase transformations were observed during tensile tests. The DSC of the heat-treated specimen shows a three-stage phase transformation process, B2→R→B19′, with temperatures shifted from negative to positive in comparison with the as-deposited one, in which transformation proceeded as follows: B2→B19′.

**Author Contributions:** Conceptualization, A.K. and O.P.; data curation, I.K.; formal analysis, O.P.; funding acquisition, A.P.; investigation, A.K.; methodology, A.K.; project administration, O.P.; resources, A.K., O.P. and A.P.; validation, A.K., O.P. and D.K.; visualization, I.K.; writing—original draft, A.K.; writing—review and editing, O.P. and D.K. All authors have read and agreed to the published version of the manuscript.

**Funding:** This research was funded by State Atomic Energy Corporation Rosatom (ROSATOM), contract No. H.4щ.241.09.20.1081, dated 4 June 2020.

**Institutional Review Board Statement:** Not applicable.

**Informed Consent Statement:** Not applicable.

**Data Availability Statement:** The data presented in this study are available upon request from the corresponding author.

**Conflicts of Interest:** The authors declare no conflict of interest.

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
