# Peer review of "Functional and Mechanical Properties of As-Deposited and Heat Treated WAAM-Built NiTi Shape-Memory Alloy"

_metals, doi:10.3390/met12061044_

Round 1
Reviewer 1 Report
The microstructure, phase composition, and mechanical properties of the NiTi shape memory alloy before and after heat treatment are compared in this study. However, it is understood that many studies dealing with WAAM NiTi shape memory alloy have been published, with all of them showing promising findings. Furthermore, heat treatment of materials is a pretty mature set of theories, and this paper's originality has yet to be reflected in anything. Therefore, it is recommended to reconsider whether to accept it after the following modifications, and some comments and suggestions are as follows.
1. “There is a group of materials that have shape memory effect. Such a group gains interest from industry and scientists trough years due to superior properties and possibility to develop new automotive, aerospace, medical, robotics applications.” should be replaced with “There is a class of materials that have shape-memory effects. Such a group has gained interest from industry and scientists over the years due to its superior properties and the possibility to develop new automotive, aerospace, medical, and robotics applications.”, and there are many other mistakes. The language needs to be revised to improve its fluency.
2. The effect of heat treatment on the microstructure and mechanical properties of materials has been widely studied, what is the significance of this paper especially highlighting to study this aspect?
3. What is the novelty of this paper compared with the previous WAAM NiTi shape memory alloy?
4. Because some figures are enormous but the labels inside are small, it will be difficult to discern the label contents, such as a and b in Figure 1, it is recommended to enlarge the numbers or letters in the figure.
5. The columnar crystal size described in Figure 2 is not consistent with the figure. The columnar crystal in zone 3 appears to be larger than that in zone 2, despite the fact that the text states that zone 3 has the smallest columnar crystal, and it is suggested that grain size be measured in different zones to provide a quantitative comparison.
Author Response
Response to Reviewer 1
Comments
The microstructure, phase composition, and mechanical properties of the NiTi shape memory alloy before and after heat treatment are compared in this study. However, it is understood that many studies dealing with WAAM NiTi shape memory alloy have been published, with all of them showing promising findings. Furthermore, heat treatment of materials is a pretty mature set of theories, and this paper's originality has yet to be reflected in anything. Therefore, it is recommended to reconsider whether to accept it after the following modifications, and some comments and suggestions are as follows.
Response 0: Greatly thank you for your qualified approach to our research, which is reflected in your comments and suggestions. Especially thank you for the comments about conceptualization. We hope we were able to clarify all the issues. All suggestions are taken into account, the information is included and/or corrected in the text.
Point 1: “There is a group of materials that have shape memory effect. Such a group gains interest from industry and scientists trough years due to superior properties and possibility to develop new automotive, aerospace, medical, robotics applications.” should be replaced with “There is a class of materials that have shape-memory effects. Such a group has gained interest from industry and scientists over the years due to its superior properties and the possibility to develop new automotive, aerospace, medical, and robotics applications.”, and there are many other mistakes. The language needs to be revised to improve its fluency.
Response 1: We've reworked the language following your advice and included all the edits in the Manuscript.
Point 2: The effect of heat treatment on the microstructure and mechanical properties of materials has been widely studied, what is the significance of this paper especially highlighting to study this aspect?
Response 2: Totally agree: we need to discuss this aspect in our article in more detail and we have already include further information in the text.
You correctly noticed that there are wide variety of research on heat treatment and its influence on the properties of NiTi alloys. But most of them are focused on the research of nearly isotropic material. From the scientific point of view, here we have anisotropic material because of manufacturing type. We believe that we have succeeded anisotropic behaviour of the deposited specimens in our work. There are much smaller number of topics focused on the influence of heat treatment on the anisotropic NiTi alloy, fabricated by wire and arc additive manufacturing.
From the industrial point of view, our work on the WAAM of NiTi with additional heat treatment for 430℃, 1h, shows the possibility to use this process for manufacturing parts with controllable transformation temperatures (for example: thermomechanical coupling). We believe that it can contribute to further implementation of this process in industrial fields.
We hope that now we clarified this aspect and have answered your question correctly.
Point 3: What is the novelty of this paper compared with the previous WAAM NiTi shape memory alloy?
Response 3: There are several aspects:
- Hight scale specimen
We are thinking that researchs in the filed of WAAM shall be focused on relaitevly hight scale printed parts. We consider the aim of the method as the reason for such statement : rapid printing of hight scale metal parts.
- We are focused on the influence of heat treatment on the hight scale specimens manufactured using WAAM-MIG technique.
You may not agree with this statement and you will be right. Nowadays there is one article (https://doi.org/10.1016/j.mtla.2021.101238), focused on the influence of heat treatment on the WAAM-MIG for NiTi SMA alloy. This fact indicates the relevance of this topic. Unfortunately, this work was not yet published at the time we were doing our research. But, even in this case we are still the only team talking about relatively hight scale specimens, deposited using WAAM-MIG process.
We have now added the article mentioned above to our reference list. Thanks again for your question.
- There are some other small aspects: we are using substrate and filler wire with the identical chemical composition instead of Ti or steel substrates compared with the other groups; the evaluation of the functional properties in our work was carried out, among other things, using a bending test.
Point 4: Because some figures are enormous but the labels inside are small, it will be difficult to discern the label contents, such as a and b in Figure 1, it is recommended to enlarge the numbers or letters in the figure.
Response 4: We have corrected the font size in the figure 1 according to your comment. In addition, other figures were checked and the letters were enlarged.
Point 5: The columnar crystal size described in Figure 2 is not consistent with the figure. The columnar crystal in zone 3 appears to be larger than that in zone 2, despite the fact that the text states that zone 3 has the smallest columnar crystal, and it is suggested that grain size be measured in different zones to provide a quantitative comparison.
Response 5: We have verified the statement. That is true, the size of the columnar crystal in zone 3 is larger than in other zones. The description has been corrected, grain size measurements have been made and included in the text. A methodology for the average grain size calculation has been added to section 2.

Reviewer 2 Report
1. Please describe the test methods and the shape of tensile test and bending specimens.
2. The transformation temperatures strongly depend on the Ni contents. Therefore, please add the Ni contents (at%) in Table 1.
Author Response
Response to Reviewer 2
Response 0: Greatly thank you for your time. Your comments and suggestions helped us complete the article with more detailed information.
Point 1: Please describe the test methods and the shape of tensile test and bending specimens.
Response 1: We have added a description of the testing methods. The information about the shapes of bending and tensile samples were also included.
Point 2: The transformation temperatures strongly depend on the Ni contents. Therefore, please add the Ni contents (at%) in Table 1.
Response 2: Table 1 has been updated with this information.

Reviewer 3 Report
The paper “1762528” related to WAAM in AM was reviewed. Please follow the comments carefully and resubmit your paper for the next consideration and reviewing process.
1. How the experiment was designed. Add more detail.
2. The explanation of section 3.3 needs to be expanded.
3. Add a short note about the results to the abstract.
4. Please add a scale bar to Figure 4.
5. How authors selected the process parameters for the experimentation?
6. Please proofread the text
7. The introduction needs to be updated by comparing the WAAM and Laser-based powder bed fusion LB-PBF which is also called SLM. Read and add the following new references.
· Evolution of temperature and residual stress behaviour in selective laser melting of 316L stainless steel across a cooling channel
· Fatigue life optimization for 17-4Ph steel produced by selective laser melting
· Proposal of design rules for improving the accuracy of selective laser melting (SLM) manufacturing using benchmarks parts
· Study of anisotropy through microscopy, internal friction and electrical resistivity measurements of Ti-6Al-4V samples fabricated by selective laser melting
8. Additive manufacturing has many advantages over the conventional manufacturing method which can be highlighted in your paper. Please read the following article and add to the introduction to show the experimental application of additive manufacturing and the advantage of this process over conventional manufacturing like machining.
Additive manufacturing a powerful tool for the aerospace industry.
Author Response
Response to Reviewer 3
Comments
The paper “1762528” related to WAAM in AM was reviewed. Please follow the comments carefully and resubmit your paper for the next consideration and reviewing process.
Response 0: Greatly thank you for your qualified approach to our research, which is reflected in your comments and suggestions. Especially thank you for the comments about the Materials and Methods section as well as the Introduction section. We hope we were able to clarify all the issues. All suggestions are taken into account, the information is included and/or corrected in the text.
Point 1: How the experiment was designed. Add more detail.
Response 1: The explanation of the experimental design has been extended.
Point 2: The explanation of section 3.3 needs to be expanded.
Response 2: We have updated section 3.3 with more details.
Point 3: Add a short note about the results to the abstract.
Response 3: A short note has been added.
Point 4: Please add a scale bar to Figure 4.
Response 4: Figure 4 has a scale bar in the lower right corner.
Point 5: How authors selected the process parameters for the experimentation?
Response 5: We have supplemented section 2 with the following information: “The process parameters were chosen to achieve maximum process stability. For this rea-son, the values were gradually changed to the best combination with minimal spatter, ef-fective gas shielding, and stable metal transfer to the weld pool.”
Point 6: Please proofread the text.
Response 6: Proofreading carried out, all corrections can now be seen in the Manuscript file.
Point 7: The introduction needs to be updated by comparing the WAAM and Laser-based powder bed fusion LB-PBF which is also called SLM. Read and add the following new references.
- Evolution of temperature and residual stress behaviour in selective laser melting of 316L stainless steel across a cooling channel
- Fatigue life optimization for 17-4Ph steel produced by selective laser melting
- Proposal of design rules for improving the accuracy of selective laser melting (SLM) manufacturing using benchmarks parts
- Study of anisotropy through microscopy, internal friction and electrical resistivity measurements of Ti-6Al-4V samples fabricated by selective laser melting
Response 7: We greatly thank you for your advice on extending the introduction by comparing these methods.
We have read all the articles you suggested. In this case, we decided to use in our literature review the latest articles devoted to the study of the SLM process using NiTi alloy, since this material is directly related to our work.
You are encouraged to read all additions in the Introduction section of the Manuscript file.
Point 8: Additive manufacturing has many advantages over the conventional manufacturing method which can be highlighted in your paper. Please read the following article and add to the introduction to show the experimental application of additive manufacturing and the advantage of this process over conventional manufacturing like machining.
- Additive manufacturing a powerful tool for the aerospace industry.
Response 8: We are totally agree with your point of view. The proposed article has been read and added to the Introduction section in accordance with your advice.

Round 2
Reviewer 1 Report
All the issues in the first round have been well addressed.
Reviewer 3 Report
This is publishable.